# Fast-Tracking Vaccine Manufacturing: CEPI’s Rapid Response Framework for the 100 Days Mission

**DOI:** 10.3390/vaccines13080849

**Published:** 2025-08-11

**Authors:** June Kim, Ramin Sabet-Azad, Dimki Patel, Gene Malin, Syed Hassan Askary, Anna Särnefält

**Affiliations:** 1Manufacturing & Supply Chain, Coalition for Epidemic Preparedness Innovations, Askekroken 11, 0277 Oslo, Norway; ramin.sabet-azad@cepi.net (R.S.-A.); dimki.patel@cepi.net (D.P.); gene.malin@cepi.net (G.M.); anna.sarnefalt@cepi.net (A.S.); 2Regulatory Affairs, Coalition for Epidemic Preparedness Innovations, Askekroken 11, 0277 Oslo, Norway; hassan.askary@cepi.net

**Keywords:** 100 Days Mission, Chemistry, Manufacturing and Controls (CMC), rapid response framework, pandemic preparedness, platform technologies, accelerated vaccine development

## Abstract

CEPI (the Coalition for Epidemic Preparedness Innovation)’s CMC rapid response framework is developed to support accelerated vaccine development, manufacturing, and roll-out for various outbreak scenarios to achieve the 100 Days Mission. The framework outlines coordinated deliverables across five functional areas: manufacturing processes, formulation, analytics, supply chain, and facilities. It could serve as a tool to streamline CMC and the related activities for rapid vaccine development, identify areas for improvement and innovation, and assess preparedness for the outbreak response. The framework emphasizes the importance of thorough preparation during interpandemic periods as the foundation for the 100 Days Mission and for gaining the confidence of health authorities and the public in vaccines.

## 1. Introduction

The 100 Days Mission is an ambitious goal to respond to the next pandemic by making safe, effective vaccines, therapeutics, and diagnostics within 100 days of recognition of a pandemic pathogen [1,2,3]. It has received support from G7, G20, and other countries [4,5,6]. It was originally outlined by CEPI in their 2022–2026 plan for epidemic and pandemic preparedness. CEPI’s 100 Days Mission particularly emphasizes vaccine readiness for initial authorization and manufacturing at scale within 100 days [7].

The public health and economic benefits of CEPI’s 100 Days Mission have been modeled under various outbreak scenarios, and it has consistently shown the potential to save millions of lives and billions of dollars [8,9,10]. The urgency of the 100 Days Mission becomes increasingly apparent as outbreaks caused by viruses continue to threaten multiple nations and regions. Examples include the recent outbreaks of Marburg in Rwanda and mpox in East and Central African nations in 2024 [11].

The rapid responses to Marburg and mpox outbreaks were catalyzed by the existing vaccine candidates ready for deployment and supported by strong coordination among the stakeholders [11,12,13,14,15,16]. For instance, over 2700 doses of an investigational vaccine against Marburg were provided by the Sabin Vaccine Institute, and the vaccination of healthcare workers in Rwanda began within 10 days of the outbreak declaration [11,12]. Similarly, vaccines approved for smallpox and mpox for adults were shipped to the affected region. The mpox vaccines received emergency use authorization (EUA) for pediatric use within 100 days of the health emergency declaration [11].

The 100 Days Mission promotes a paradigm shift in vaccine development and manufacturing for emergencies. It emphasizes the importance of preparation during interpandemic periods, and there have been important initiatives and significant investments to facilitate preparedness for the next pandemic [1,3].

In 2024, the WHO published an R&D Blueprint with updated priority and prototype pathogens that have significant potential to cause the next pandemic. Priority pathogens are well-characterized pathogens with significant public health emergency threats. Lassa fever virus (Arenaviridae) and Nipah virus (Paramyxoviridae) are a couple of examples of priority pathogens [17]. As a result, developing vaccine candidates against the priority pathogens has been prioritized, and a few candidates have already progressed into clinical trials [18].

Prototype pathogens are model pathogens representing viral families with high pandemic risk. Significant investments have been made to develop novel antigens, evaluate their proof-of-concept immunity, and build a vaccine library with prototype as well as priority pathogens. This library could provide vaccine candidates against pathogens of concern once the novel pathogen is sequenced and mapped against the priority and prototype pathogens, enabling acceleration of vaccine manufacturing and roll-out [19,20,21].

Another key enabler for the 100 Days Mission is the vaccine platform technology. A platform technology is a well-characterized and reproducible technology with accumulated prior knowledge and experience in medical countermeasure (MCM) development [22]. It is disease-agnostic and results in plug-and-play use and streamlined regulatory processes [11]. Therefore, it could allow significant acceleration of initial CMC activities such as process and analytical development, analytical qualification by leveraging the platform experience, early manufacturing, and ultimately end-to-end development of novel vaccines [23,24].

Emergency use authorization (EUA) is a conditional approval that allows access to an unlicensed MCM or unapproved uses of a licensed MCM during public health emergencies. The EUA process needs to be rapid but rigorous, as the regulatory guidance for EUA indicates that the benefits of an MCM should outweigh the risks and require data supporting safety and effectiveness. Clinical trials are expected to continue to obtain additional safety and effectiveness information after EUA is obtained and pursue licensure [25,26,27,28].

Justifying the benefits of novel vaccines within 100 days is challenging. However, this can be facilitated by employing a vaccine candidate with established proof-of-concept immunity from the vaccine library, if appropriate, and well-characterized platform technologies that consistently produce safe and effective vaccines. These optimized platform technologies could enhance regulatory and public confidence in the novel vaccines, which are developed with an extremely accelerated development timeline. Vaccine developers could integrate the novel antigen into their platform and expedite the development and manufacturing of the vaccine candidate by leveraging prior knowledge. Therefore, it is crucial to continue investing pandemic preparedness, focusing on key technical areas that could achieve the 100 Days Mission.

In this paper, we discuss CEPI’s CMC rapid response framework. This framework is designed for accelerated scenarios including 100 days, where multiple CMC and related activities are conducted almost simultaneously. It requires a thorough understanding of the necessary tasks when an outbreak occurs and/or a public health emergency is declared. The framework provides tailored CMC checklists for pandemic preparedness and key deliverables for accelerated vaccine development, manufacturing, and roll-out. These are divided into five technical development areas: (1) drug substance (DS) and drug product (DP) process, (2) formulation and stability, (3) analytical methods, (4) material controls and supply chain, and (5) manufacturing and facilities. We propose using this framework as guidance to streamline CMC and related activities during pandemic preparation and outbreak response phases. Additionally, we anticipate that the framework could serve vaccine developers as a tool to identify areas for improvement and innovation for rapid response and as a benchmark to assess preparedness levels.

## 2. Rapid Response Scenarios

The rapid response framework addresses three different outbreak scenarios, as described in Figure 1. Each scenario illustrates a varying level of epidemiology risk, preparedness, and familiarity with the pathogen. The response times to EUA vary between 100 days and 230 days, which are simulated with consideration of the current time-limiting steps of manufacturing, clinical trials, and regulatory processes for the respective scenarios. We anticipate that they could be further accelerated through innovations and fine-tuning of vaccine development processes, ultimately leading to 100 days in all scenarios.

The first scenario begins with a well-understood vaccine candidate with known safety and dosing information. This situation arises when the pathogen of concern closely resembles priority or prototype pathogens for which doses and safety profiles of potential vaccine candidates are available. It allows the existing vaccine candidate to be adapted for the specific outbreak pathogen, and phase 2/3 trials could be initiated immediately. This is the scenario in which the 100 Days Mission is most likely to be achieved.

The second scenario describes an outbreak where the pathogen of concern is related to the priority and prototype pathogens designated in the WHO R&D Blueprint [17]. A lead antigen is identified for vaccine development from a vaccine library, but it has limited safety and immunogenicity information. A phase 1/2 trial is necessary to establish safety and dosing; therefore, it could take longer than the first scenario, with an estimated timeline to EUA of 150 to 180 days at least.

The third scenario involves an outbreak of an unknown disease (disease X), where the pathogen is unrelated to the priority and prototype pathogens designated in the WHO R&D Blueprint [17]. In this scenario, the response should commence with antigen design, followed by preclinical studies and a phase 1 trial. It could take a minimum of 200 to 230 days to receive EUA. This is the theoretically shortest possible duration with current practices and is still an acceleration compared to the COVID-19 vaccine, which obtained EUA in approximately one year [3].

The CMC rapid responses are simulated according to the three scenarios (Figure 1). As soon as a public health emergency is declared, the immediate CMC reaction is to manufacture and release clinical trial materials (CTMs) for a phase 3 trial in the 100-day scenario, or for a phase 1/2 trial in the 150–180-day scenario. Similarly, the 200–230-day scenario requires the manufacturing of phase 1/2 CTMs soon after completing antigen design (Figure 1b).

Concurrent with the manufacturing of CTM batches, another immediate response involves scaling out to regional manufacturing to enhance access to vaccine candidates in the affected regions. This can be achieved through existing partnerships with local manufacturers who share the same platform technology or through de novo technology transfer to manufacturers who do not have experience with the technology, which requires more time and technical support. The former approach is better suited for an accelerated scenario.

In addition, CMC responses must be coordinated with regulatory and clinical trial activities to enable an accelerated and seamless vaccine roll-out, ultimately achieving EUA within the defined timelines. It is also important to emphasize that acceleration strategies for clinical trials are being developed to support rapid evidence generation for safety and effectiveness, such as through immunobridging or single-dose event-based trials, as noted in Figure 1a [29,30]. It is critical to understand the clinical trial needs and timing, as these will influence manufacturing scale, schedule, and the number of batches.

## 3. CMC Rapid Response Framework

### 3.1. Preparation During Interpandemic Periods

Accelerated vaccine development within 100 days of identification of the outbreak pathogen can only be achieved when vaccine developers are ready to execute the simulated scenarios, as shown in Figure 1b. The rapid response framework underscores CMC strategies for pandemic preparedness, and key activities and deliverables deemed necessary for acceleration are described in Table 1.

The development of platform technologies is emphasized in each technical area for preparation (Table 1). As vaccine development continues for either routine or emerging pathogens during interpandemic periods, there are tremendous opportunities to develop, optimize, and mature platform technologies throughout the vaccine development lifecycle. This results in the accumulation of development and manufacturing experience using the platform technologies. This prior knowledge could streamline the best practices across CMC areas, including technology transfer and manufacturing, and it could accelerate vaccine development processes [22,31].

It is also important to reflect on lessons learned from the COVID-19 pandemic. There were numerous regulatory challenges due to the unprecedented speed of making billions of vaccine doses available globally. Process validation (PV), comparability of numerous batches manufactured by different manufacturers, and stability requirements for regulatory approval are only a few examples [32,33,34]. The observed regulatory challenges could be mitigated by establishing platform technologies for the next pandemic response. Vaccine developers need to demonstrate the reproducible performance of platform technologies with multiple vaccine candidates and consider taking the vaccine candidates to licensure, if appropriate. This is likely to enhance operational flexibility and streamline the regulatory pathway when the same platforms are applied for rapid vaccine development [22,28]. It is crucial to seek regulatory feedback from health authorities (HAs) and assess risks related to platform technologies throughout the lifecycle of vaccine development.

As platform technologies become mature, they define raw materials, consumables, and starting materials for manufacturing. Appropriate suppliers could be identified, qualified, and reserved for both routine operations, such as conventional vaccine manufacturing during interpandemic periods, and rapid response under emergency. Additionally, manufacturing facilities and quality control (QC) laboratories need to be qualified for technology transfer, production, and release of vaccine candidates. The routine operations can facilitate the continued supply of key materials and training on platform technologies, with benefits multiplied in low- and middle-income countries (LMICs). This could result in workforce development and quality system maturity in these regions. Strengthening global manufacturing capabilities is critical for pandemic preparedness and is achievable through continuous investment [35,36,37,38,39,40,41].

Several platforms have been adopted for vaccine development, including mRNA, viral vectors, and protein subunits. mRNA vaccines are well-suited for rapid response and can be updated quickly with new antigens [42]. However, they are temperature-labile and require careful formulation development for storage conditions suitable for equitable access [43]. Viral vector vaccines can trigger strong immune responses and provide long-lasting immunity. Viral vectors can be revised quickly against new pathogens, but their manufacturing processes tend to be complex [44]. Protein subunit vaccines are made of viral proteins or their fragments as antigens, which could elicit immune responses. These technologies are not as adaptable to new pathogens as mRNA and viral vectors and often need adjuvants to boost immunity [45,46].

There are conserved properties across vaccine candidates against different pathogens within the same platform, which include certain critical quality attributes (CQAs) and quality control strategies. Some developers may prefer to use the same dosage form and administration route. These product-specific characteristics could be extended to platform characteristics based on the accumulated data. Some of the QTPP elements could be identified as platform characteristics, and this can increase the efficiency of vaccine development by eliminating the need to re-justify these characteristics with new vaccine candidates using the same platform.

The long-term stability and in-use stability of different vaccine candidates within the same platform can be highly consistent when the same formulation, dosage form, and storage conditions are applied. Although some stability-indicating attributes may present different trends for each vaccine, the stability data for each attribute can be aggregated from multiple vaccines to build stability models. These models can help to predict shelf-lives of novel vaccine candidates within the same platforms [47], potentially accelerating the regulatory process by eliminating the need to wait for stability data required for the regulatory filing.

A good example of how platforms can accelerate the timeline is found in monoclonal antibody (mAb) therapies developed for COVID-19 treatment. The development of several monoclonal antibodies was accelerated by 75% or more, reducing the timeline to clinical trials. This acceleration was possible due to mature mAb platform processes, manufacturing capabilities available in many regions, and regulatory experience, which were based on the accumulated prior knowledge from the approval of more than 100 monoclonal antibodies in the past [48].

Innovating manufacturing technologies is essential to accomplish the 100 Days Mission in any outbreak scenario and enhance equitable access. Key innovations need to be developed and applied to vaccine development and manufacturing before the need arises for acceleration. Integrating platform technologies with digital tools and modeling could facilitate process understanding, advance process controls, and lead to successful manufacturing at a rapid pace [49,50]. Developing thermostable formulations and stability modeling could increase shelf-life, support rapid shelf-life prediction and definition, and improve vaccine access globally [23,51]. Developing alternative adjuvants with increased immune response and comparable safety profiles to conventional ones and generating evidence for their platform potential could enhance access to critical excipients for vaccine development [52].

It is also vital for innovators to engage regulators to assess compliance risks of novel technologies. This may require early adoption of the technologies for vaccine manufacturing at risk. This proactive approach can promote optimization, familiarize regulators with the new technologies, and help avoid significant delays in the regulatory process during rapid responses.

Preparation of investigational reserves (or stockpiles) of vaccine candidates against pathogens with pandemic potential could offer an advantage of prompt responses to outbreaks of related pathogens. The Marburg outbreak mentioned above is an example of the positive impact of investigational stockpiles on outbreak control [11].

**Table 1 vaccines-13-00849-t001:** CMC and other technical deliverables for preparation during the interpandemic period.

Category	DS/DP Process	Formulation and Stability	Analytical	Material Controls and Supply Chain	Manufacturing Facilities
Expected Requirement	Platform manufacturing processes established and discussed with HAs	Platform formulation (including, e.g., adjuvants) established, consulted with HAs	Platform analytical methods (in-process, release, stability, and characterization) established for DS/DP	Platform raw materials and consumables defined (Bill of Materials)	GMP clinical and commercial manufacturing site(s) for DS/DP in the region(s) identified and qualified
	^1^→ Platform manufacturing process characterization with vaccine candidates	Platform DP presentation defined	Platform (master) release specifications established for DS/DP	→ Raw material and consumables’ suitability/origins defined	→ Contract in place for access in emergency
	→ CQAs, CPPs, PAR, and NOR defined based on platform experience	→ Adjuvants compatibility with various vaccine candidates demonstrated	→ Platform assays and data discussed with HAs	→ Multicompendial-grade (high-quality) raw material availability ensured	QC labs in region(s) identified and qualified
	→ Data consistency from PV using platform processes	→ DS/DP stability for platform demonstrated	Phase-appropriate validation for platform assays performed	→ Risk assessment performed for critical raw materials	→ Contract in place for access in an emergency
	→ Market authorization obtained with a minimal number of vaccine candidates if feasible	→ In-use stability for platform demonstrated	Develop vaccine-specific assays (such as potency and ID assays) for prototype, exemplar vaccines/disease X library, if appropriate	→ Critical material attributes defined	Site(s) in compliance with applicable HA regulations
	→ Platform QTPP identified	→ Accelerated stability for platform conducted	Rapid release methods developed and validated in phase-appropriate manner	Platform starting materials defined	Platform manufacturing process and analytical methods transferred
	→ Viral clearance validated for platform, if appropriate	→ Primary packaging validated (CCIT, dose recovery, leachable/extractables) with various vaccine candidates	Analytical tech transfer protocol developed and executed with at least one vaccine candidate	→ Release specifications established	→ Facility fit assessed
	Establish platform tech transfer	Shelf-life determined with platform formulation	Comparability strategy discussed/established	→ Origin of starting materials defined	→ Risk assessment initiated
	→ Platform manufacturing process description, MBR and other documentation required for GMP manufacturing, appropriate for any clinical phase and commercialization	Risk registers developed for platform and/or vaccine candidate-specific formulation	HA consultation for rapid response analytical strategy	Adjuvants identified and suppliers established, if applicable	Live fire drill to confirm successful execution of various process steps, if applicable
	→ Tech transfer strategy and protocol developed for platform and executed, if appropriate	Train workforce to develop SME	Analytical lifecycle management strategy in place	Analytical reagents defined and suppliers identified	Train workforce and develop SME
	Develop stable cell lines and/or master cell banks of priority and/or prototype vaccines/disease X library, if appropriate	Rolling QA review process established	Risk registers developed for platform and/or vaccine candidate-specific analytical methods	License agreement in place, if required	Rolling QA review process established
	Develop viral stocks of priority and prototype pathogens, if applicable		Train workforce to develop SME	Supply chain defined and qualified per CMC framework—supply chain and COGS [53]	
	Comparability strategy discussed with HAs and established		Rolling QA review process established	Risk registers of raw materials for platform- and product-specific development	
	→ Platform database prepared with comparability between engineering, phase 1/2a, and other CTM batches for vaccine candidates			Alternative raw material suppliers identified and qualified, based on risk registers	
	IND/IMPD and BLA templates developed			→ Identify and qualify multiple suppliers of critical raw materials	
	Develop risk-based product lifecycle plans			→ Maintain inventory necessary for rapid response	
	→ Manage platform risk assessment			Train workforce to develop SME	
	→ Continue platform optimization			Rolling QA review process established	
	Train workforce to develop SME				
	Rolling QA review process established				
Additional Considerations	Process innovation for 100 Days Mission—speed, for example	Formulation innovation—thermal stability, for example	Analytical innovation—high throughput, automation, etc.	Primary packaging defined and validated	Perform process validation using the platform to gain PV experience
	Develop prototype vaccines/disease X library using platform process	Evaluate adjuvants from adjuvant library [52]	Identify and qualify analytical CROs for vaccine-specific analytical method development and characterization	Shipping validation strategy in place and conducted, if appropriate	Optimize processes per live fire drill experience, if appropriate
	Develop manufacturing strategies with alternative technologies to SCL/MCB	Develop and validate stability model, if appropriate			
	→ Comparability assessed between alternative technologies (such as transient) and conventional processes using SCL/MCB				
	Investigational reserve/stockpile of vaccine candidates prepared, if appropriate				

^1^ The symbol → and increased indentation describes additional deliverables for the activity listed above. Abbreviations: biologics license applications (BLAs), clinical trial material (CTM), container closure integrity test (CCIT), contract research organization (CRO), cost of good (COG), critical process parameter (CPP), identity (ID), investigational medicinal product dossier (IMPD), investigational new drug (IND), master batch record (MBR), master cell bank (MCB), normal operating range (NOR), proven acceptable range (PAR), quality assurance (QA), quality control (QC), quality target product profile (QTPP), stable cell line (SCL), subject matter expert (SME).

### 3.2. The 100-Day Scenario—Outbreak of a Familiar Pathogen and Vaccine Candidates Available with Known Safety and Dosing Information

The 100-day scenario represents the most probable simulation to deliver the 100 Days Mission. It starts with vaccine candidates that have established safety and dosing information when the pathogen is identified and a public health emergency of international concern is declared (Figure 1). In the rapid response scenario, we anticipate that these vaccine candidates are developed using platform technologies and ready for scale-up and scale-out.

As shown in Figure 1b, the immediate CMC response involves manufacturing vaccine candidates at scale and releasing them as early as 6 weeks, considering the 100 Days Mission. At-scale manufacturing should supply a sufficient number of vaccine doses for populations affected by an outbreak pathogen as quickly as possible but should be determined based on available manufacturing capacities and the vaccine roll-out plan under each outbreak situation. The clinical trial materials (CTMs) are intended to support phase 2b/3 clinical trials. The expected CMC deliverables and parallel activities for and during the phase 2b/3 manufacturing are described in Table 2.

**Table 2 vaccines-13-00849-t002:** The 100-day scenario—deliverables for CTM manufacturing for phase 2b/3.

Category	DS/DP Process	Formulation and Stability	Analytical	Material Controls and Supply Chain	Manufacturing and Facilities
Expected Requirement	Manufacturing DS/DP using platform process at scale, yielding sufficient doses for phase 2b/3	Initiate DS/DP stability of phase 2b/3 CTM	QTPP finalized for the vaccine candidate	Raw material supplies secured for manufacturing	CTM manufacturing site(s) identified and slots secured
	→ Manufacture WCB/WVS, in consideration of manufacturing billions of doses in case of pandemic response	Conduct accelerated stability of phase 2b/3 CTM	Platform analytical fit with the vaccine candidate demonstrated	→ Critical or special/single-source raw materials identified and supply secured	Site(s) in compliance with applicable HA regulations
	DS/DP released using platform analytical release and specification	Shelf-life established with the vaccine candidate, based on phase 1 batch or platform formulation, if appropriate	Specification justified for the vaccine candidate	Cold chain storage and shipping strategy to clinical sites established	→ Site(s) inspected and GMP-certified by local authority(ies) for vaccine candidate manufacturing
	Continue consultation with HAs for IND approval/PV strategy	Risk registers developed for vaccine candidate-specific formulation	Reference standard strategy in place	Risk registers developed for product-specific raw or starting materials	QC lab(s) identified and resources secured
	Comparability demonstrated with previous batches		Analytical comparability with previous batches evaluated		
	Comparability protocol in place (for phase 2b/3, PV, commercial, and tech-transferred batches)		Comparability protocol in place (for phase 2b/3, PV, commercial and tech-transferred batches)		
	Platform process fit with the vaccine candidate demonstrated		Develop and qualify vaccine-specific analytical methods, if necessary		
	→ Platform viral clearance verified, if appropriate		Analytical validation of critical tests		
	IND/IMPD for phase 2b/3 submitted		Continue consultation with HA analytical validation strategy for conditional approval		
	Tech transfer to regional manufacturers, if appropriate		Analytical tech transfer to regional manufacturers, if necessary		
	→ Train workforce for tech transfer SME		Risk registers developed for vaccine candidate-specific analytical methods		
	Risk registers developed, specific for vaccine candidate				
Additional Considerations	COGS confirmed with the vaccine candidate	In-use stability demonstrated with the vaccine candidate, if appropriate		Raw material batch-to-batch comparability evaluated	
	If platform fit is not demonstrated at scale, optimize and update process and control strategies for vaccine candidate	Validate the stability model with vaccine candidate-specific stability data			

Abbreviations: working cell bank (WCB); working virus seed (WVS).

While the expected activities and deliverables in this scenario are similar to those in conventional vaccine development, the timeline is significantly compressed and vaccine-specific development activities are included. The scenario assumes that there are manufacturing facilities and QC labs that are experienced in the platform processes and analytics, stocked with raw materials, maintained with a quality system suitable for good manufacturing practice (GMP) operations, and ready to initiate the campaign immediately. Additionally, this scenario can be achieved if there are risk-based comparability protocols, platform technology master files (PTMFs), or alternative regulatory pathways for rapid review aligned and approved by HAs. There are seasoned workforces to conduct CMC, regulatory, and clinical activities required for the conditional approval, as discussed in preparation for rapid response. There are also vaccine-specific characteristics that might require adaptation and/or development of vaccine-specific process parameters and analytical methods, including potency.

Upon completion of the CTM manufacturing at scale and release, analytical and process validations may follow within an extremely aggressive timeline, such as 8 weeks (Table 3). In this scenario, we anticipate the best case of preparedness where developers have gained significant experience with platform technologies during the interpandemic periods, the platform fit is demonstrated during the large-scale manufacturing, and vaccine-specific development is conducted. There should be alignment with HAs to conduct a risk-based PV exercise for conditional approval, with a commitment to fulfill PV requirements for licensure or post-licensure if necessary [54].

**Table 3 vaccines-13-00849-t003:** The 100-day scenario—process and analytical validation for conditional approval.

Category	DS/DP Process	Formulation and Stability	Analytical	Material Controls and Supply Chain	Manufacturing and Facilities
Expected Requirement	Conducting PV per HA consultation	Continue CTM stability testing and demonstrate alignment with platform stability experience	Complete analytical validation per HA consultation	Raw material supplies secured for process validation	Site(s) in compliance with applicable HA regulations
	Manufacturing consistency of the vaccine candidate demonstrated	Stability of validation batches	Continue consultation with HA for conditional and marketing authorization	→ Raw material stocks established suitable for process validation per risk registers	
	Continue consultation with HA for conditional approval strategy and post-conditional approval commitments	Conduct accelerated stability of selected validation batches	Continue risk assessment for vaccine candidate-specific analytical methods	Cold chain storage and shipping strategy for global supply established	
	Application for conditional approval submitted	Extend shelf-life with the vaccine candidate per stability results		Secondary packaging developed	
	Continue risk assessment for vaccine candidate	Continue risk assessment for vaccine candidate-specific formulation			
Additional Considerations	Tech transfer to regional manufacturers, if appropriate		Tech transfer to NCLs	Raw material batch-to-batch comparability evaluated	Identify and secure slots at regional manufacturers, if appropriate
				Alternative raw material suppliers identified and qualified, if appropriate	

Abbreviations: national control laboratory (NCL).

Scale-out to additional manufacturing facilities, especially in affected regions, should be planned at a minimum or initiated to enable equitable access to the vaccines (Table 2 and Table 3). As discussed in Preparation During Interpandemic Periods, it would be critical to have qualified manufacturing facilities and QC labs with experience with platform technologies in regions and alignment in comparability criteria between pre- and post-tech transfer batches. As comparability assessment is performed with vaccine-specific manufacturing batches, employing additional characterization tests beyond platform analytical release tests is necessary to be sure of comparability over the lifecycle of the vaccine and, therefore, consistent safety and effectiveness of commercial batches [54]. Without such preparedness, there would be significant delays in identifying regional manufacturers, scheduling the slots, tech transfer, manufacturing, and following regulatory filing. In addition, close collaboration of regional national control laboratories (NCLs) would be necessary for the necessary technology transfer of release assays for independent control testing [55].

Although the goal of the 100 DM is to obtain conditional approval, we must consider the next 100 days to obtain market authorization for the vaccine candidates. It is crucial to complete the post-conditional approval commitment to gain public confidence in the quality and safety of the vaccines (Table 4).

**Table 4 vaccines-13-00849-t004:** The 100-day scenario—post-conditional approval commitment for marketing authorization (the second 100 days).

Category	DS/DP Process	Formulation and Stability	Analytical	Material Controls and Supply Chain	Manufacturing and Facilities
Expected Requirement	Continue PV to meet HA commitment	Stability continued and aligned with platform stability experience	Complete commitment to post-conditional approval	Supply chain strategy for post-marketing authorization commitment defined	Site(s) in compliance with applicable HA regulations
	Continue consultation with HA for post-marketing authorization commitment	Revisit shelf-life with vaccine for possible extension	Consult HA for post-marketing authorization commitment (such as updating specification/criteria, testing methods, etc.)	Shipping validation (product-specific)	Pre-approval inspection (PAI) completed
	Process optimization and list of changes submitted per the post-approval commitment	Continue risk assessment for vaccine candidate-specific formulation	Tech transfer to NCLs	Secondary packaging is updated and ready for commercialization	Fulfill HA commitments for conditional approval
	Continue risk assessment for vaccine		Continue risk assessment for vaccine-specific analytical methods		Consult with HA for post-marketing authorization commitment for the manufacturing sites and QC laboratories
Additional Considerations				Alternative raw material suppliers identified and qualified, if appropriate	

### 3.3. The 150–180-Day Scenario—Outbreak of a Pathogen Similar to a Prototype and Vaccine Candidates Available with Limited Safety and Dosing Information

The 150–180-day scenario simulates a situation where a lead antigen has already been developed for the pathogen of concern, and its proof-of-concept immunogenicity is demonstrated in vitro or in vivo when an outbreak occurs and a public health emergency is declared. The antigen could be part of the prototype vaccine library developed per the WHO R&D Blueprint, and the pathogen of concern is related to one of the prototype pathogens [17].

Figure 1b shows that the immediate CMC response involves good manufacturing practice (GMP) manufacturing for phase 1/2a of a vaccine candidate at risk by inserting the antigen sequence into an established and suitable platform vaccine modality [42,44,45]. This will expedite the clinical trial to gain safety and dose information.

It may be necessary to have good laboratory practice (GLP) batches manufactured in parallel to the GMP batches, if regional regulators require preclinical and toxicology studies. The parallel GLP manufacturing could also help to generate additional data for the platform fit assessment of the novel antigen with the platform process, formulation, and analytical methods. However, it is worth noting that an approach to accelerate the drug safety evaluation process has been initiated by certain health authorities with reduced animal testing requirements but with scientifically validated new approach methodologies. This will drive acceleration and also cost reduction for vaccine development in routine operations and emergencies by eliminating the need for additional GLP manufacturing [56,57].

In this scenario, the expected timeline for conditional approval could extend to 150–180 days, mainly due to the need for a phase 1/2a clinical trial to gain safety and dose information. The expected activities and deliverables for GLP and GMP manufacturing for phase 1/2a are described in Table 5 and Table 6, respectively. Continuous consultation with HAs and proactive risk management in all listed categories are critical to ensure successful manufacturing, clinical trials, and regulatory approval.

In parallel to the phase 1/2a trial, GMP manufacturing for phase 2b/3 could be conducted at risk, and the vaccine development continues to proceed with conditional and market approval according to the 100-day scenario.

**Table 5 vaccines-13-00849-t005:** The 150–180-day scenario—deliverables for GLP manufacturing for preclinical evidence generation and toxicology.

Category	DS/DP Process	Formulation and Stability	Analytical	Material Controls and Supply Chain	Manufacturing and Facilities
Expected Requirement	GLP manufacturing using platform manufacturing process	Compatibility of platform formulation and adjuvant demonstrated antigen, if necessary	QTPP for vaccine candidate proposed, in consideration of TPP	Raw materials used for GLP manufacturing representative of GMP raw materials	Vaccine developer(s) and pilot facility(ies) identified
	Platform fit with antigen demonstrated	GLP material real-time stability and accelerated stability initiated	Analytical platform fit with vaccine candidate demonstrated	Risk registers developed for product-specific raw materials	
	Initiate tech transfer to GMP facility for CTM manufacturing	Risk registers developed for vaccine candidate-specific formulation	Platform IPC and IPM justified with the vaccine candidate		
	Initiate consultation with HAs for IND and preclinical/clinical strategies (pre-IND consultation)		Platform release specification justified with the vaccine candidate		
	Develop a lifecycle management plan for vaccine candidate		Develop and qualify vaccine-specific analytical methods		
	Risk assessment developed for vaccine candidate		Analytical tech transfer to QC lab(s)		
			Analytical lifecycle management strategy for the vaccine candidate in place		
			Preliminary reference standard in place for CTM release		
			Risk registers developed for vaccine candidate-specific analytical methods		
Additional Considerations	If platform fit is not achieved, optimize and update process and control strategies for vaccine candidate		Optimize analytical methods, if platform fit is not justified		
	Initiate cell line development, if necessary		Optimize IPC or IPM, if platform fit is not justified		
	Master cell bank manufactured/released, if necessary				

Abbreviations: in-process control (IPC), in-process measurement (IPM), target product profile (TPP).

**Table 6 vaccines-13-00849-t006:** The 150–180-day scenario—deliverables for manufacturing for phase 1/2a.

Category	DS/DP Process	Formulation and Stability	Analytical	Material Controls and Supply Chain	Manufacturing and Facilities
Expected Requirement	Manufacturing DS/DP using platform process, yielding sufficient doses for phase 1/2a	Initiate DS/DP stability of phase 1a/2 CTM	Continue to develop and qualify vaccine-specific methods, if necessary	Raw material supplies secured for manufacturing	CTM manufacturing site(s) identified
	→ Establish/test MCB and WCB in parallel at risk, if necessary	Conduct accelerated stability of phase 1/2a CTM	Reference standard strategy in place	→ Critical or special/single-source raw materials identified and supply secured	Site(s) in compliance with applicable HA regulations
	→ Establish/test MVS and WVS in parallel at risk, if necessary	Shelf-life established with the vaccine candidate	Tox material shown to be representative of phase 1/2a CTM	Cold chain storage and shipping strategy to clinical sites established	→ Site(s) inspected and GMP certified by local authority(ies) for vaccine candidate manufacturing
	DS/DP released using platform analytical release and specification	Continue risk assessment for vaccine candidate-specific formulation	Analytical comparability strategy and protocol in place (for phase1/2a and later GMP batches)		QC lab(s) identified
	Comparability protocol in place		Update QTPP for the vaccine candidate, if necessary		
	Continue consultation with HAs for IND approval/clinical trial strategy		Continue risk assessment for vaccine candidate-specific analytical		
	IND/IMPD for phase 1/2a submitted				
	Continue risk assessment for vaccine candidate				
Additional Considerations	Tech transfer to regional manufacturers, if necessary			Alternative raw material suppliers identified and qualified, if necessary	

### 3.4. The 200–230-Day Scenario—Unknown Pathogen and No Vaccine Candidate Available

The 200–230-day scenario simulates the situation of an unknown pathogen (disease X), necessitating a rapid response that begins with a novel antigen design. Pathogen identification through sequencing is required before proceeding with the antigen design.

There have been significant research and development efforts for the rapid development of novel antigens by leveraging artificial intelligence (AI). AI and high-throughput screening tools can increase the probability of success for safe and immunogenic antigen development in a compressed timeline [19,46,50]. It is possible that multiple antigens could advance to preclinical proof-of-concept or even phase 1/2a clinical trials until a lead candidate is secured.

As shown in Table 7, it is critical to select an appropriate platform for the novel antivaccine, ensuring its fit for the established manufacturing and release platform technologies. At the same time, phase-appropriate manufacturing and QC facilities, along with raw material suppliers, need to be secured for upcoming campaigns based on previous experience and risk assessment, as discussed in Preparation During Interpandemic Periods. The subsequent development would follow the process outlined in the 100-day and 150–180-day scenarios, with the timeline for the conditional approval extended to 200–230 days due to the added step for antigen design.

It is our ambition to accelerate the conditional approval of novel vaccines toward 100 days in any outbreak scenario. As mentioned in earlier sections, the acceleration requires innovation and fine-tuning of development processes. It also requires science- and risk-based decision-making process to progress quickly to the next steps.

For example, the 200–230-day response is not rapid enough under an acute disease X outbreak scenario, and there may be a higher risk tolerance for accelerating vaccine development even further. Accumulated knowledge on priority pathogens and vaccine library development, empowered by AI, could assist in down-selecting lead antigens in silico. This could lead to immediate CTM manufacturing at risk, and the proof-of-concept immunity could be assessed in parallel to the manufacturing. This scenario leads to 150–180 days for EUA.

**Table 7 vaccines-13-00849-t007:** The 200–230-day scenario—deliverables from antigen design phase.

Category	DS/DP Process	Formulation and Stability	Analytical	Material Controls and Supply Chain	Manufacturing and Facilities
Expected Requirement	Choose a suitable platform for vaccine candidate	Platform formulation identified	Platform analytical methods identified	Raw materials and consumables required for manufacturing identified	Phase 1/2a CTM manufacturing sites identified
	Rapid gene synthesis and antigen expression executed	Formulation compatibility with vaccine candidates evaluated	Platform analytical method fit assessed	Suppliers identified	Phase 2b/3 CTM and commercial manufacturing sites in screening
	High-throughput in vitro screening tools in place	Platform stability fit assessed.	Initiate vaccine candidate-specific assay development	Supply chain risk assessment initiated	Manufacturing and tech transfer risk assessment initiated
	Lab-scale production for manufacturability, platform fit assessment, and PoC animal study	→Vaccine candidate stability study initiated (real-time and accelerated conditions) in parallel	Risk registers developed for vaccine candidate-specific analytical	Purchase orders issued	QC lab(s) identified or in screening
	Risk assessment of vaccine development initiated	Risk registers developed for vaccine candidate-specific formulation			

## 4. Discussion

The rapid response framework discussed in this paper highlights numerous parallel and/or staggered activities across CMC and related functional areas for the ambitious 100 Days Mission, which require effective coordination to meet key regulatory milestones, such as IND/IMPD approval for clinical trials, conditional approval, and market authorization.

This framework also encourages developers to innovate and optimize platforms through vaccine lifecycle management, ensuring safety, quality, speed, and equitable access. Continuous consultation with HAs is emphasized to gain regulators’ confidence in platform technologies and vaccines during routine and accelerated vaccine development scenarios.

Gaps are identified and lessons learned are collected from recent outbreak cases such as Ebola Zaire, mpox, Marburg, and COVID-19 [11,15,24,32,33]. The experiences provide insights into how to streamline vaccine development and manufacturing more effectively. The CMC rapid response framework is a work in progress. It will be updated with best practices as CEPI and its partners continue to pave the way toward the 100 Days Mission and be shared on the website “CEPI Technical Resources (https://cepi-tr.tghn.org)”.

## 5. Conclusions

The CMC rapid response framework simulates how the 100 Days Mission can be delivered in public health emergencies and also highlights the current gaps in CMC and related areas in achieving this ambitious goal. The accelerated scenarios presented in this paper require a paradigm shift in vaccine development frameworks and stress the importance of being prepared by developing vaccine platform technologies, generating evidence of benefits, and managing risks during interpandemic periods. It discusses critical initiatives catalyzing the paradigm shift and addressing the identified gaps, such as manufacturing technology innovations and maturing regional manufacturing capabilities, for rapid and equitable access to vaccines. It also considers the importance of synergistic coordination across multidisciplinary areas, including CMC, clinical, and regulatory, ultimately to deliver the 100 Days Mission in public health emergencies with all pathogens.

## Figures and Tables

**Figure 1 vaccines-13-00849-f001:**
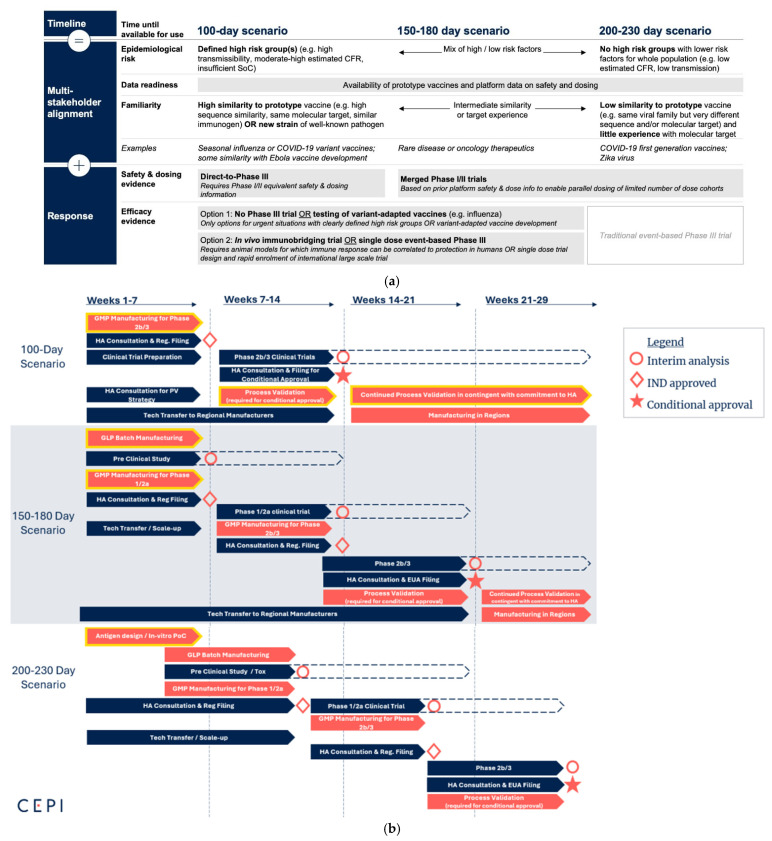
Outbreak scenarios with varying lengths of responses. (**a**) The speed of vaccine development depends on several factors, including vaccine candidates with known safety and effectiveness and the severity and risks of the situation. Reprint from Delivering Pandemic Vaccines in 100 Days [1]; (**b**) Rapid response scenarios: 100-day scenario—Outbreak of familiar pathogen and vaccine candidates available with known safety and dosing information; 150–180-day scenario—Pathogen similar to prototype and vaccine candidates available with limited safety and dosing information; and 200–230-day scenario—Unknown pathogen and no vaccine candidate available. CMC lists for the yellow-framed activities are described in Tables 2–7 under representative scenarios. Abbreviations: case fatality rate (CFR), standard of care (SoC), good laboratory practice (GLP), good manufacturing practice (GMP), health authority (HA), process validation (PV), proof of concept (PoC).

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
