# Peer review of "Fast-Tracking Vaccine Manufacturing: CEPI’s Rapid Response Framework for the 100 Days Mission"

_vaccines, 2025, doi:10.3390/vaccines13080849_

Round 1
Reviewer 1 Report
Comments and Suggestions for Authors
Kim and colleagues present CEPI’s framework of CMC deliverables for a number of scenarios/ contexts relevant to outbreak preparedness and response. This is a valuable piece of work and certainly merits publication. I have two major comments – one regarding format, and the other regarding ‘scenario 3’.
Format
The key content is essentially a set of ‘to do lists’, presented in the format of seven large tables – these are the point of novelty, with the text outside the tables really playing a supporting / explanatory role – and so the way in which this key content is presented will be important for the reader. I think the authors face a challenge, however, in that the lists don’t ‘fit’ particularly well into the format of a PDF manuscript, or indeed into tables. The table contents are lists organised in five parallel columns, but there is no commonality between items displayed horizontally adjacently in rows. Currently there is one table for pre-outbreak preparation, three relating to sub-elements of ‘scenario 1’, two relating to sub-elements of ‘scenario 2’, and one relating to ‘scenario 3’. Some of the content is repeated across tables, but it’s not easy to cross-reference which deliverables may contribute to more than one scenario, or to compare how activities might differ between scenarios. Table 1, in particular, is hard to follow as it extends across four pages.
I wonder whether it may be more helpful if the authors could consider alternative presentations of the tabular information (which need not require change to the content, or the body of the text).
It’s tricky to know what might ‘work’ better than the current format. One possibility might be to use instead five tables themed around each of the five areas covered by the current columns. Rows could represent linked sub-topics (for example, for a ‘materials & supply chain’ table, there might be rows on raw material procurement, cold chain / shipping, packaging etc) – possibly with sub-rows representing more specific tasks which recur across scenarios. Seven columns could then represent each of the scenarios / sub-elements covered by the seven current tables, indicating which activity within each sub-topic is required in each scenario.
A link to the same information in editable Excel format may also be helpful in allowing readers to re-order the information as relevant to their own needs.
I think would also be helpful if the explanation of the scenarios provided in the current Figure 1 & 2 could indicate more clearly how these have been used to structure the sets of deliverables. Firstly, it would be helpful for the 100-day, 150-180-day and 200-230-day scenarios to be labelled as scenarios 1-3 respectively in Figure 1, as this is the terminology used in Figure 2 and the rest of the paper. Secondly, it may be helpful for Figures 1 & 2 to be presented adjacent to each other (i.e. to become panels A/B within a single combined Figure 1). Legibility of the current Figure 2 could be improved, and indications added to show how scenarios 1 & 2 are each sub-divided into different sub-sets of deliverables (for example draw boxes around the elements included in each of the sub-scenarios).
Scenario 3
230 days for a scenario akin to the development of first-gen COVID-19 vaccines seems to me to be somewhat lacking in ambition, as this is only saving ~33% vs the times actually achieved for COVID-19.
Even event-based efficacy studies can be very small / fast if the attack rate is high (i.e. the exposure is common in the study population).
First-gen COVID-19 vaccine development did not face major problems of antigen design / platform suitability (which are the focus of Scenario 3). Instead the initial designs, guided by prior experience with the pathogen family, were adequate – but vaccine deployment was slowed by the lack of pre-pandemic CMC platform preparedness, and the fact that programmes that vaccinated thousands of individuals in May / June 2020 did so in the context of lockdowns (such that exposure was too rare for a rapid efficacy readout).
Arguably it’s becoming less likely that the world will encounter a scenario in which it would seem that delivering an obvious antigen design using an existing vaccine platform would have such a low chance of ‘useful success’ (e.g. efficacy against death, even if imperfect) that it would not be worth pursuing this ‘obvious’ route as ‘plan A’. I wonder if the authors might therefore consider a more ambitious / optimistic parallelised approach for a COVID-19-like scenario i.e. fastest possible development with best-guess antigen design and existing CMC platform, with exploration of novel antigen designs / platforms (scenario 3) happening in parallel as a ‘belt and braces’. An approach like the current scenario 3 would, in my view, only be appropriate alone in exceptional circumstances, with a clearly ‘immunologically very difficult’ pathogen (e.g. something like HIV spread by the respiratory route).
Minor comments
Line 61 – should be ‘A platform technology is a well-characterized …’
Line 68-70 – ‘The regulatory guidance for EUA indicates that the benefits of MCM should outweigh the risks, necessitating a rigorous EUA process comparable to the conventional regulatory process’. While clearly it is correct that the EUA process is / should be rigorous, it is not identical to that for conventional authorisation. It would be more useful to explain ways in which they differ than to imply that they are so similar they are effectively identical.
Legend to Figure 1 – the abbreviations SoC and CFR should be spelled out, or avoided.
Figure 1, ‘efficacy option 2’ – immunobridging vs an animal model & a single-dose event-based Phase III are very different – should these actually be option 2 & option 3? The latter is not fundamentally different from a traditional phase III so is really an intermediate option
Figure 2 is hard to read. ‘Manf’ is used as an abbreviation but not explained
The depiction of tech transfer to regional manufacturers happening in parallel with product development in Figure 2 perhaps suggests that this is de novo tech transfer to a facility with no prior experience of the platform. In reality maximum speed is only likely to be achieved if platform processes have been transferred ahead of the outbreak – as alluded to in the text and Table 1. Perhaps this could be clearer.
Lines 240-249 & Fig 2b mention a ‘GLP batch’ as a core requirement. OECD & WHO GLP guidance is explicit that it is applicable only to non-clinical safety evaluation, and use of the terminology in other contexts is discouraged. Lines 246-249 state that approaches have been initated ‘with reduced animal testing requirements but with scientifically validated new methodologies’. It would be worth being clearer about this – some HAs are definitely willing to accept FIH studies being started without non-clin safety data, on the basis of platform experience. Moreover, a truly aggressive approach to reaching FIH quickly would provide a GMP batch as quickly as a GLP batch, with the difference perhaps being only that animal studies could be started before the completion of GMP release testing. Beyond the start of FIH, there may be a need for animal data prior to EUA, but GMP material will be available. Can a ‘GLP batch’ therefore be removed from the critical path?
It would be helpful to be clearer about the objective of defining these lists of deliverables. Are they primarily to provide a ready-made list of what would need to be done in different scenarios, as a starting point to be used in an actual emergency? Or are they to guide pre-outbreak planning, and strengthening of platform readiness?
Author Response
Please see the attachment.
Comments 1: The key content is essentially a set of ‘to do lists’, presented in the format of seven large tables – these are the point of novelty, with the text outside the tables really playing a supporting / explanatory role – and so the way in which this key content is presented will be important for the reader. I think the authors face a challenge, however, in that the lists don’t ‘fit’ particularly well into the format of a PDF manuscript, or indeed into tables. The table contents are lists organised in five parallel columns, but there is no commonality between items displayed horizontally adjacently in rows. Currently there is one table for pre-outbreak preparation, three relating to sub-elements of ‘scenario 1’, two relating to sub-elements of ‘scenario 2’, and one relating to ‘scenario 3’. Some of the content is repeated across tables, but it’s not easy to cross-reference which deliverables may contribute to more than one scenario, or to compare how activities might differ between scenarios. Table 1, in particular, is hard to follow as it extends across four pages. I wonder whether it may be more helpful if the authors could consider alternative presentations of the tabular information (which need not require change to the content, or the body of the text). It’s tricky to know what might ‘work’ better than the current format. One possibility might be to use instead five tables themed around each of the five areas covered by the current columns. Rows could represent linked sub-topics (for example, for a ‘materials & supply chain’ table, there might be rows on raw material procurement, cold chain / shipping, packaging etc) – possibly with sub-rows representing more specific tasks which recur across scenarios. Seven columns could then represent each of the scenarios / sub-elements covered by the seven current tables, indicating which activity within each sub-topic is required in each scenario. A link to the same information in editable Excel format may also be helpful in allowing readers to re-order the information as relevant to their own needs.
|
Response 1: We appreciate your thoughtful comments and recommendations. Initially, we started with the five table themes you proposed. However, we encountered challenges with the "preparation", as it includes a longer list of activities. Consequently, all five tables would span multiple pages. Additionally, fitting seven columns on one page, even in landscape orientation, proved difficult. Given that the texts are structured per scenario, the seven table themes seem to work better. As authors explain in line 401-404, the framework will be posted on the website “CEPI Technical Resources (https://cepi-tr.tghn.org)”. We will prepare the file with five table themes as you proposed.
|
Comments 2: I think would also be helpful if the explanation of the scenarios provided in the current Figure 1 & 2 could indicate more clearly how these have been used to structure the sets of deliverables. Firstly, it would be helpful for the 100-day, 150-180-day and 200-230-day scenarios to be labelled as scenarios 1-3 respectively in Figure 1, as this is the terminology used in Figure 2 and the rest of the paper. Response 2: As Figure 1 is a reprint, the legends cannot be updated. Instead, we updated Scenario 1 – 3 to 100-day, 150-180 days, and 200-230 days scenarios throughout the manuscript, including Figure 1(b).
|
Comments 3: Secondly, it may be helpful for Figures 1 & 2 to be presented adjacent to each other (i.e. to become panels A/B within a single combined Figure 1). Legibility of the current Figure 2 could be improved, and indications added to show how scenarios 1 & 2 are each sub-divided into different sub-sets of deliverables (for example draw boxes around the elements included in each of the sub-scenarios). |
Response 3: We appreciate your thoughtful comments. We have changed the figure 2 as you proposed. Now it is Figure 1b, clarity is improved, and yellow frames are placed in the boxes which are discussed separately in tables.
Comments 4: 230 days for a scenario akin to the development of first-gen COVID-19 vaccines seems to me to be somewhat lacking in ambition, as this is only saving ~33% vs the times actually achieved for COVID-19. Even event-based efficacy studies can be very small / fast if the attack rate is high (i.e. the exposure is common in the study population). First-gen COVID-19 vaccine development did not face major problems of antigen design / platform suitability (which are the focus of Scenario 3). Instead the initial designs, guided by prior experience with the pathogen family, were adequate – but vaccine deployment was slowed by the lack of pre-pandemic CMC platform preparedness, and the fact that programmes that vaccinated thousands of individuals in May / June 2020 did so in the context of lockdowns (such that exposure was too rare for a rapid efficacy readout). Arguably it’s becoming less likely that the world will encounter a scenario in which it would seem that delivering an obvious antigen design using an existing vaccine platform would have such a low chance of ‘useful success’ (e.g. efficacy against death, even if imperfect) that it would not be worth pursuing this ‘obvious’ route as ‘plan A’. I wonder if the authors might therefore consider a more ambitious / optimistic parallelised approach for a COVID-19-like scenario i.e. fastest possible development with best-guess antigen design and existing CMC platform, with exploration of novel antigen designs / platforms (scenario 3) happening in parallel as a ‘belt and braces’. An approach like the current scenario 3 would, in my view, only be appropriate alone in exceptional circumstances, with a clearly ‘immunologically very difficult’ pathogen (e.g. something like HIV spread by the respiratory route). Response 4: We appreciate your comments and the challenge for us to develop better scenario. We agree and a paragraph is included to address your comments in line 378-387. “It is our ambition to accelerate the conditional approval of novel vaccines toward 100 days in any outbreak scenarios. As mentioned in earlier sections, the acceleration requires innovation and fine-tuning of development processes. It also requires science- and risk-based decision-making process to progress quickly to the next steps. For example, the 200-230 day response is not rapid enough under an acute disease X outbreak scenario, and there may be a higher risk tolerance for accelerating vaccine development even further. Accumulated knowledge in priority pathogens and vaccine library development, empowered by AI, could assist in down-selecting lead antigens in silico. This could lead to immediate CTM manufacturing at risk, and the proof-of-concept immunity could be assessed in parallel to the manufacturing. This scenario leads to 150-180 days for EUA.”
Comments 5: Line 61 – should be ‘A platform technology is a well-characterized …’ Response 5: Corrected in line 59.
Comments 6: Line 68-70 – ‘The regulatory guidance for EUA indicates that the benefits of MCM should outweigh the risks, necessitating a rigorous EUA process comparable to the conventional regulatory process’. While clearly it is correct that the EUA process is / should be rigorous, it is not identical to that for conventional authorisation. It would be more useful to explain ways in which they differ than to imply that they are so similar they are effectively identical. Response 5: We agree with your comments. The paragraph is revised. Line 67-72: Emergency use authorization (EUA) is a conditional approval that allows access to unlicensed MCM or unapproved uses of licensed MCM during public health emergencies. EUA process needs to be rapid but rigorous, as the regulatory guidance for EUA indicates that the benefits of MCM should outweigh the risks and require data supporting safety and effectiveness. It is expected to continue clinical trials to obtain additional safety and effectiveness information after EUA is obtained and pursue licensure [25-28].
Comments 6: Legend to Figure 1 – the abbreviations SoC and CFR should be spelled out, or avoided. Response 6: The abbreviations are included in the figure 1.
Comments 7: Figure 1, ‘efficacy option 2’ – immunobridging vs an animal model & a single-dose event-based Phase III are very different – should these actually be option 2 & option 3? The latter is not fundamentally different from a traditional phase III so is really an intermediate option Response 7: Figure 1(a) is a reprint from CEPI 100 days report published in Nov 2022. The underlying assumption for 200-230 days was that it was not different from current clinical practices, therefore, it was grayed out as Traditional event-based Phase III trial. Please consider that this is not the area authors can update for this manuscript.
Comments 8: Figure 2 is hard to read. ‘Manf’ is used as an abbreviation but not explained Response 8: Figure 2 is changed to Figure 1(b). Clarity and Manf are updated according to reviewer’s comment.
Comments 9: The depiction of tech transfer to regional manufacturers happening in parallel with product development in Figure 2 perhaps suggests that this is de novo tech transfer to a facility with no prior experience of the platform. In reality maximum speed is only likely to be achieved if platform processes have been transferred ahead of the outbreak – as alluded to in the text and Table 1. Perhaps this could be clearer. Response 9: We agree with reviewer’s comments. The paragraph is revised. “Line 159-650: Concurrent with the manufacturing CTM batches, another immediate response involves scaling out to regional manufacturing to enhance access to vaccine candidates in the affected regions. This can be achieved through existing partnerships with local manufacturers who share the same platform technology, or through de novo technology transfer to manufacturers who do not have experience with the technology, which requires more time and technical support. The former approach is better suited for an accelerated scenario. “
Comments 10: Lines 240-249 & Fig 2b mention a ‘GLP batch’ as a core requirement. OECD & WHO GLP guidance is explicit that it is applicable only to non-clinical safety evaluation, and use of the terminology in other contexts is discouraged. Lines 246-249 state that approaches have been initated ‘with reduced animal testing requirements but with scientifically validated new methodologies’. It would be worth being clearer about this – some HAs are definitely willing to accept FIH studies being started without non-clin safety data, on the basis of platform experience. Moreover, a truly aggressive approach to reaching FIH quickly would provide a GMP batch as quickly as a GLP batch, with the difference perhaps being only that animal studies could be started before the completion of GMP release testing. Beyond the start of FIH, there may be a need for animal data prior to EUA, but GMP material will be available. Can a ‘GLP batch’ therefore be removed from the critical path? Response 10: We revised the paragraphs according to your comments. Line 332-345: “Figure 1(b) shows that the immediate CMC response involves good manufacturing practice (GMP) manufacturing for phase 1/2a of a vaccine candidate at risk, by inserting the antigen sequence into an established and suitable platform vaccine modality [42, 44, 45]. It will expedite the clinical trial to gain safety and dose information. It may be necessary to have good laboratory practice (GLP) batches to be manufactured in parallel to the GMP batches, if regional regulators require preclinical and toxicology studies. The parallel GLP manufacturing could also help to generate additional data for the platform fit assessment of the novel antigen with the platform process, formulation, and analytical methods. However, it is worth noting that an approach to accelerate the drug safety evaluation process has been initiated by certain health authorities with reduced animal testing requirements but with scientifically validated new approach methodologies. It will drive acceleration and also cost reduction for vaccine development in routine and emergencies by eliminating the need of additional GLP manufacturing [56,57]. “
Comments 11: It would be helpful to be clearer about the objective of defining these lists of deliverables. Are they primarily to provide a ready-made list of what would need to be done in different scenarios, as a starting point to be used in an actual emergency? Or are they to guide pre-outbreak planning, and strengthening of platform readiness? Response 11: We revised the paragraphs according to your comments. Line 83-95: “In this paper, we discuss CEPI’s CMC rapid response framework. This framework is designed for accelerated scenarios including 100 days, where multiple CMC and related activities are conducted almost simultaneously. It requires a thorough understanding of the necessary tasks when an outbreak occurs and/or a public health emergency is declared. The framework provides tailored CMC checklists for pandemic preparedness and key deliverables for accelerated vaccine development, manufacturing, and roll-out. These are divided into five technical development areas: (1) drug substance (DS) and drug product (DP) process, (2) formulation and stability, (3) analytical methods, (4) material controls and supply chain, and (5) manufacturing and facilities. We propose using this framework as guidance to streamline CMC and related activities during pandemic preparation and outbreak response phases. Additionally, we anticipate that the framework could serve vaccine developers as a tool to identify areas for improvement and innovation for rapid response, and as a benchmark to assess preparedness levels.”
|

Reviewer 2 Report
Comments and Suggestions for Authors
The paper "Fast-tracking vaccine manufacturing: CEPI's rapid response framework for 100 days mission" provides an insightful overview of CMC activities and opportunities in different health emergency scenarios. Overall, the topic fits with Vaccines scope and I recommend publication, provided that the following considerations are taken into account.
1- Lines 70-72: the authors highlight the importance of established platform technologies for consistent quality of safe and efficacious vaccines over time. The driver for this statement in the 100 days context should be clarified (why consistent quality over time and not the rapid response to new pathogens?).
2- Figure 1 and lines 115-122: in 100 day scenario, starting directly with Ph3 is very difficult unless option 1 in Figure 1 is considered (emerging new virus variants); do the author have examples of option 2 ? Assuming that there are no safety concerns, established correlate of protection could indeed enable rapid Ph3 start, and it would be helpful to add some considerations on how this could be achieved (e.g., appropriate nonclinical models, use of challenge studies, ...) and what are the challenges associated to such a fast Ph3 start (e.g. dose identification or population size, which could be pathogen- specific and might not be fully predicted). Mention of adaptive clinical protocols allowing adjustment of study depending on interim outcomes could be helpful- despite this paper focus is mostly on CMC aspects, some short text on clinical strategy could be very relevant to provide key context for CMC considerations.
3- The paper does not cover a specific vaccine type and aims at defining type- agnostic technical deliverables. Nevertheless, it would be important to clarify the extent of use of platform depends on the vaccine type, independently on the maturity of the manufacturer/ platform.
4- Table 1: platform QTPP is mentioned: by definition QTPP also includes product- specific considerations, depending on the pathogen, the chosen presentation to support a given population etc. This should be clarified.
5- Table 1: DS/ DP stability for platform: stability trend is product-specific (even within mRNA vaccines, with significant platform elements, the pathogen and consequent length of the construct have an impact on stability). What is the meaning of DS/ DP stability for platform and how is it demonstrated?
5- Table 1: a reference is made to development and validation of stability modeling, but it is recommended to mention establishment of stability modeling approaches for specific CQAs/ vaccine types; also, there is a recommendation to evaluate adjuvants from CEPI adjuvant library- this seems to be self- promotional and not necessarily exhaustive; suggest to mention CEPI adjuvant library as an example and not as the exclusive reference
6- Lines 158-160: it could be helpful to mention learnings also from platform use for mAbs used during COVID crisis, for two main reasons: (i) some principles could be similar in the CMC space, as reported in literature (see for instance AAPS J. 2022 Sep 27;24(6):101, doi: 10.1208/s12248-022-00751-9); (ii) pandemic situations could be addressed considering both therapeutics and vaccines as done at the beginning of the COVID crisis.
7- Table 3: Tech transfer to NCLs is mentioned. Is it worth considering a simplification/ harmonization of the NCL approach and tests choice especially in case of 100-days scenarios? Also, reliance shared Analytical Target Profile (ICH Q14) to support analytical bridging and flexible choice of technologies across NCLs and manufacturers would be helpful.
8- Would it be appropriate to mention deferral of some process and analytical validation activities, with post- approval commitment of completion? Also, more emphasis on the importance of analytical characterization to support comparability is critical: even in case of platform product and process, some product- specific adaptation may be needed, to an extent intrinsically dependent on vaccine type. For this reason, relying only on platform technology to support comparability is not systematically possible. More information on both PV strategies and comparability are reported in tech-brief_april-2021_regulation-of-covid-19-vaccines_synopsis_-aug2020_feb2021.pdf
Author Response
Please see the attchement. The changes made per reviewer 2's comments are highlighted in green in the updated manuscript.
Comments 1:Lines 70-72: the authors highlight the importance of established platform technologies for consistent quality of safe and efficacious vaccines over time. The driver for this statement in the 100 days context should be clarified (why consistent quality over time and not the rapid response to new pathogens?).
|
Response 1: The paragraph is revised to clarify platform technologies. “Line 73-78: Justifying the benefits of novel vaccines within 100 days is challenging. However, this can be facilitated by employing a vaccine candidate with established proof-of-concept immunity from the vaccine library, if appropriate, and well-characterized platform technologies that consistently produce safe and effective vaccines. These optimized platform technologies could enhance regulatory and public confidence in the novel vaccines, which are developed with an extremely accelerated development timeline.”
|
Comments 2: Figure 1 and lines 115-122: in 100 day scenario, starting directly with Ph3 is very difficult unless option 1 in Figure 1 is considered (emerging new virus variants); do the author have examples of option 2 ? Assuming that there are no safety concerns, established correlate of protection could indeed enable rapid Ph3 start, and it would be helpful to add some considerations on how this could be achieved (e.g., appropriate nonclinical models, use of challenge studies, ...) and what are the challenges associated to such a fast Ph3 start (e.g. dose identification or population size, which could be pathogen- specific and might not be fully predicted). Mention of adaptive clinical protocols allowing adjustment of study depending on interim outcomes could be helpful- despite this paper focus is mostly on CMC aspects, some short text on clinical strategy could be very relevant to provide key context for CMC considerations.
Response 2: We appreciate your comments. For the rapid response frameworks, three specific scenarios were considered. The first scenario involves starting a phase 3 trial immediately, provided that safety and dose information are available, which align with WHO priority pathogens. We agree that it is critical to understand clinical strategies to manufacture clinical trial materials at an appropriate scale and time. Therefore, we added a paragraph in line 166 -172 to address reviewer’s comment.
Lines166-172: “In addition, CMC responses must be coordinated with regulatory and clinical trial activities to enable an accelerated and seamless vaccine roll-out, ultimately achieving EUA within the defined timelines. It is also important to emphasize that acceleration strategies for clinical trials are being developed to support rapid evidence generation for safety and effectiveness, such as through immunobridging or single dose event-based trials, as commented in Figure 1(a) [29,30]. It is critical to understand the clinical trial needs and timing, as it would influence manufacturing scale, schedule, and the number of batches.”
|
Comments 3: The paper does not cover a specific vaccine type and aims at defining type- agnostic technical deliverables. Nevertheless, it would be important to clarify the extent of use of platform depends on the vaccine type, independently on the maturity of the manufacturer/ platform. |
Response 3: We agree with your comment. A paragraph was added in line 211-220. Line 211-220: Several platforms have been adopted for vaccine development, including mRNA, viral vectors, and protein subunits. mRNA vaccines are well-suited for rapid response and can be updated quickly with new antigens [42]. However, they are temperature-labile and require careful formulation development for storage conditions suitable for equitable access [43]. Viral vector vaccines can trigger strong immune responses and provide long-lasting immunity. Viral vectors can be revised quickly against new pathogens, but their manufacturing processes tend to be complex [44]. Protein subunit vaccines are made of viral proteins or their fragments as antigens, which could elicit immune responses. These technologies are not as adaptable to new pathogens as mRNA and viral vectors and often need adjuvants to boost immunity [45,46].
Comments 4: Table 1: platform QTPP is mentioned: by definition QTPP also includes product- specific considerations, depending on the pathogen, the chosen presentation to support a given population etc. This should be clarified.
Response 4: We agree with your comment that there are product-specific considerations of QTPP. We also believe that there are certain elements of QTPP could be considered platform if data support. We added a new paragraph to clarify platform QTPP in line 221 – 228. We also soften the language in Table 1 to “Platform QTPP identified” instead of “developed.” Line 221 – 228: There are conserved properties across vaccine candidates against different pathogens within the same platform, which include certain critical quality attributes (CQAs) and quality control strategies. Some developers may prefer to use the same dosage form and administration route. These product-specific characteristics could be extended to platform characteristics, based on the accumulated data. Some of QTPP elements could be identified as platform, and this can increase efficiency of vaccine development by eliminating the need to re-justify these characteristics with new vaccine candidates using the same platform.
Comments 5: DS/ DP stability for platform: stability trend is product-specific (even within mRNA vaccines, with significant platform elements, the pathogen and consequent length of the construct have an impact on stability). What is the meaning of DS/ DP stability for platform and how is it demonstrated? Comments 6: Table 1: a reference is made to development and validation of stability modeling, but it is recommended to mention establishment of stability modeling approaches for specific CQAs/ vaccine types; also, there is a recommendation to evaluate adjuvants from CEPI adjuvant library- this seems to be self- promotional and not necessarily exhaustive; suggest to mention CEPI adjuvant library as an example and not as the exclusive reference
Response 5 & 6: As comment 5 and 6 comment on stability, we would like to address both comments. We understand that “platform stability” needs to be clarified, and therefore, we added a new paragraph line 229-236.
Line 229-236: The long-term stability and in-use stability of different vaccine candidates within the same platform can be highly consistent when the same formulation, dosage form, and storage conditions are applied. Although some stability-indicating attributes may present different trends for each vaccine, the stability data for each attribute can be aggregated from multiple vaccines to build stability models. These models can help to predict shelf-lives of novel vaccine candidates within the same platforms [47], potentially accelerating regulatory process by eliminating the need to wait for stability data required for the regulatory filing.
Regarding the adjuvant library – Authors were asked to include a discussion regarding the adjuvant library at the time of writing this article. We intend to keep this unless it interrupts the discussion. Instead, we revised CEPI adjuvant library to adjuvant library in Table 1.
Comments 7: Lines 158-160: it could be helpful to mention learnings also from platform use for mAbs used during COVID crisis, for two main reasons: (i) some principles could be similar in the CMC space, as reported in literature (see for instance AAPS J. 2022 Sep 27;24(6):101, doi: 10.1208/s12248-022-00751-9); (ii) pandemic situations could be addressed considering both therapeutics and vaccines as done at the beginning of the COVID crisis.
Response 7: Authors agree with the recommendation. A new paragraph was added in line 237-243. We used the reference https://doi.org/10.1016/j.copbio.2022.102798, instead. Line 237-243: A good example of how platforms can accelerate the timeline is found in monoclonal antibody (mAb) therapies developed for COVID-19 treatment. The development of several monoclonal antibodies was accelerated by 75% or more, reducing the timeline to clinical trials. This acceleration was possible due to mature mAb platform processes, manufacturing capabilities available in many regions, and regulatory experience, which were based on the accumulated prior knowledge from the approval of more than 100 monoclonal antibodies in the past [48].
Comments 8: Table 3: Tech transfer to NCLs is mentioned. Is it worth considering a simplification/ harmonization of the NCL approach and tests choice especially in case of 100-days scenarios? Also, reliance shared Analytical Target Profile (ICH Q14) to support analytical bridging and flexible choice of technologies across NCLs and manufacturers would be helpful.
Response 8: Authors appreciate the comments. While we recognize the value of harmonizing NCL approaches and establishing analytical bridging across NCLs, these efforts fall outside the current scope of this article. Our primary aim is to raise awareness among vaccine developers regarding essential CMC activities for pandemic preparedness and rapid response. Within this context, we mention that there could be a need to engage with NRAs and NCL. Although a more streamlined NCL framework would be helpful, implementing such changes would require broad, collaborative action beyond the capacity of any single vaccine developer or manufacturer.
Comments 9: Would it be appropriate to mention deferral of some process and analytical validation activities, with post- approval commitment of completion? Also, more emphasis on the importance of analytical characterization to support comparability is critical: even in case of platform product and process, some product- specific adaptation may be needed, to an extent intrinsically dependent on vaccine type. For this reason, relying only on platform technology to support comparability is not systematically possible. More information on both PV strategies and comparability are reported in tech-brief_april-2021_regulation-of-covid-19-vaccines_synopsis_-aug2020_feb2021.pdf
Response 9: We agree with your comments. Thank you for sharing the reference. We made revision in line 306-308 and line 314-317, based on the reference provided.
Line 306-308: There should be alignment with HAs to conduct a risk-based PV exercise for conditional approval, with a commitment to fulfill PV requirements for licensure or post-licensure if necessary [54]. Line 314-317: As comparability assessment is performed with vaccine-specific manufacturing batches, employing additional characterization tests beyond platform analytical release tests is necessary to be sure of comparability over the lifecycle of the vaccine and, therefore, consistent safety and effectiveness of commercial batches [54].
|

Reviewer 3 Report
Comments and Suggestions for Authors
The manuscript outlines CEPI’s thoughts on how to fast track vaccine manufacturing in the event of an emergency. This is an ongoing issue related to global preparedness in terms of vaccine manufacturing capacity and the subject warrants attention in the literature. The manuscript is centered around CEPI’s goal of being able to respond to a pandemic within 100 days of recognition of the pandemic pathogen.
- Lines 31-36: This is somewhat misleading. The progress in the case of Marburg and mpox virus was only due to existing development programs, to include a vaccine that was already approved for use in the case of mpox by some regulators, with a sizable inventory. Similarly, a limited amount of clinical trial material was already available for Marburg and doses were only used under a clinical trial. This was more a matter of availability and it isn’t clear if the authors expedited manufacturing in these cases. The authors make this clear in lines 39-41, but it isn’t clear if or how manufacturing was accelerated in the face of outbreaks, or if it was just a matter of products being available due to existing programs. The example of proactive investments in the following paragraphs do a much better job of laying out how manufacturing efforts have been, and will be, accelerated.
- Lines 60-65: This paragraph may benefit from another sentence or two specifically focused on how platform technologies may facilitate early manufacturing activities (e.g. process development, assay development/qualification, etc.).
- Figure 1: This is not critical for the figure itself, but it may worth noting in the text how CEPI’s overall portfolio aligns with these scenarios. For example, with CEPI’s Lassa vaccine investments, based on maturity of the portfolio, which scenario might that currently fall in if a large epidemic were to occur?
- Line 160: Identifying suppliers and securing facilities/workforce should not be understated, that will be a significant gap anywhere, and perhaps more so in LMIC. The note of reservation for routine operations and rapid response is critical, but additional details in terms of thinking of what the routine operations would entail would be helpful.
- Lines 166-174: The need for innovation in manufacturing is clear, but this paragraph would be better suited if it stressed the importance of investments in innovation prior to the need to meet the 100 day response. Inclusion of innovation in manufacturing, assuming that means regulatory interactions on the approach may currently be limited, has potential to result in significant regulatory delays.
- General comment: In any cases of pathogens of pandemic potential that do NOT yet have licensed products, do the authors have any thoughts on advancing programs to regulatory approvals? Those would be situations where the 100 day mission would be met.
- Line 213: At-scale manufacturing may require definition. That may be very different pathogen-by-pathogen. If pandemic level capacity is assumed in all cases, there are very few viral families where at-scale manufacturing can be assumed.
- Lines 217-219: Agree that the goal is critical, but additional caveats should be included in terms of schedule impact and regulatory considerations when moving into new facilities that may not have experience with a platform.
Author Response
Please see the attachment.
Any changes made per reviewer 3's comments are highlighted in blue in the updated manuscript.
Comments 1: Lines 31-36: This is somewhat misleading. The progress in the case of Marburg and mpox virus was only due to existing development programs, to include a vaccine that was already approved for use in the case of mpox by some regulators, with a sizable inventory. Similarly, a limited amount of clinical trial material was already available for Marburg and doses were only used under a clinical trial. This was more a matter of availability and it isn’t clear if the authors expedited manufacturing in these cases. The authors make this clear in lines 39-41, but it isn’t clear if or how manufacturing was accelerated in the face of outbreaks, or if it was just a matter of products being available due to existing programs. The example of proactive investments in the following paragraphs do a much better job of laying out how manufacturing efforts have been, and will be, accelerated.
|
Response 1: We appreciate your comment. We agree with the feedback and revised the section accordingly. Line 30 – 36: Urgency for the 100 days mission becomes increasingly apparent as outbreaks caused by viruses continue to threaten multiple nations and regions. The examples include the recent outbreaks of Marburg in Rwanda and mpox in East and Central African nations in 2024 [11]. The rapid responses to Marburg and mpox outbreaks were catalyzed by the existing vaccine candidates ready for deployment and supported by strong coordination among the stakeholders [11-16].
|
Comments 2: Lines 60-65: This paragraph may benefit from another sentence or two specifically focused on how platform technologies may facilitate early manufacturing activities (e.g. process development, assay development/qualification, etc.).
Response 2: We revised the sentence based on your comment. Line 63 – 66: Therefore, it could allow significant acceleration of initial CMC activities such as process and analytical development, analytical qualification by leveraging the platform experience, early manufacturing, and ultimately end-to-end development of novel vaccines [23,24].
|
Comments 3: Figure 1: This is not critical for the figure itself, but it may worth noting in the text how CEPI’s overall portfolio aligns with these scenarios. For example, with CEPI’s Lassa vaccine investments, based on maturity of the portfolio, which scenario might that currently fall in if a large epidemic were to occur? |
Response 3: We appreciate your comment. While the CEPI portfolio is advancing to meet the outlined scenarios, we believe that discussion of this progress falls outside the scope of this manuscript and may divert from the main topic. Reviewer 2 also noted that mentioning the CEPI adjuvant library could be perceived as self-promotional. We acknowledge this concern and suggest that there will be future opportunities for CEPI to publish updates on the portfolio’s progress.
Comments 4: Line 160: Identifying suppliers and securing facilities/workforce should not be understated, that will be a significant gap anywhere, and perhaps more so in LMIC. The note of reservation for routine operations and rapid response is critical, but additional details in terms of thinking of what the routine operations would entail would be helpful.
Response 4: We revised paragraph according to your comment. Line 200 – 210: As platform technologies become mature, they define raw materials, consumables, and starting materials for manufacturing. Appropriate suppliers could be identified, qualified, and reserved for both routine operations, such as conventional vaccine manufacturing during interpandemic periods, and rapid response under emergency. Additionally, manufacturing facilities and quality control (QC) laboratories need to be qualified with technology transfer, production, and release of vaccine candidates. The routine operations can facilitate the continued supply of key materials and training on platform technologies, with benefits multiplied in low- and middle-income countries (LMICs). This could result in workforce development and quality system maturity in these regions. Strengthening global manufacturing capabilities is critical for pandemic preparedness and is achievable through continuous investment [35-41].
Comments 5: Lines 166-174: The need for innovation in manufacturing is clear, but this paragraph would be better suited if it stressed the importance of investments in innovation prior to the need to meet the 100 day response. Inclusion of innovation in manufacturing, assuming that means regulatory interactions on the approach may currently be limited, has potential to result in significant regulatory delays.
Response 5: We revised the content to address your comment in line 244-247 and line 255-259. Line 244-247: Innovating manufacturing technologies is essential to accomplish the 100 days mission in any outbreak scenario and enhance equitable access. Key innovations need to be developed and applied to vaccine development and manufacturing before their needs arise for acceleration. Line 255-259: It is also vital for innovators to engage regulators to assess compliance risks of novel technologies. It may require early adoption of the technologies for vaccine manufacturing at risk. This proactive approach can promote optimization, familiarize regulators with the new technologies, and help avoid significant delays in the regulatory process during rapid responses.
Comments 6: General comment: In any cases of pathogens of pandemic potential that do NOT yet have licensed products, do the authors have any thoughts on advancing programs to regulatory approvals? Those would be situations where the 100 day mission would be met.
Response 6: We appreciate your comment. CEPI considers “next 100 day” after the 100 days mission is met, and it includes to take vaccine candidate to licensure. For CEPI portfolio, we also consider end-to-end development and support until commercialization ready. Line 322 – 325 mentions about the next 100 days: Although the goal of the 100 DM is to obtain conditional approval, we must consider the next 100 days to obtain market authorization for the vaccine candidates. It is crucial to complete the post-conditional approval commitment to gain public confidence in the quality and safety of the vaccines (Table 4).
Comments 7: Line 213: At-scale manufacturing may require definition. That may be very different pathogen-by-pathogen. If pandemic level capacity is assumed in all cases, there are very few viral families where at-scale manufacturing can be assumed.
Response 7: We added a sentence describing at-scale manufacturing in line 282-285. Line 282-285: At-scale manufacturing should supply a sufficient number of vaccine doses for populations affected by an outbreak pathogen as quickly as possible, but should be determined based on available manufacturing capacities and vaccine roll-out plan under each outbreak situation.
Comments 8: Agree that the goal is critical, but additional caveats should be included in terms of schedule impact and regulatory considerations when moving into new facilities that may not have experience with a platform.
Response 8: We agree with your comment and, therefore, we emphasized tech transfer to facilities as part of preparation and those can be found in following lines. Line 159-165: Concurrent with the manufacturing CTM batches, another immediate response involves scaling out to regional manufacturing to enhance access to vaccine candidates in the affected regions. This can be achieved through existing partnerships with local manufacturers who share the same platform technology, or through de novo technology transfer to manufacturers who do not have experience with the technology, which requires more time and technical support. The former approach is better suited for an accelerated scenario. Line 200 – 210: As platform technologies become mature, they define raw materials, consumables, and starting materials for manufacturing. Appropriate suppliers could be identified, qualified, and reserved for both routine operations, such as conventional vaccine manufacturing during interpandemic periods, and rapid response under emergency. Additionally, manufacturing facilities and quality control (QC) laboratories need to be qualified with technology transfer, production, and release of vaccine candidates. The routine operations can facilitate the continued supply of key materials and training on platform technologies, with benefits multiplied in low- and middle-income countries (LMICs). This could result in workforce development and quality system maturity in these regions. Strengthening global manufacturing capabilities is critical for pandemic preparedness and is achievable through continuous investment [35-41].
In addition, we added a sentence to reflect on your comment in Line 317-319: Without such preparedness, it would result in significant delays in identifying regional manufacturers, scheduling the slots, tech transfer, manufacturing, and following regulatory filing.
|

Round 2
Reviewer 1 Report
Comments and Suggestions for Authors
The revisions have addressed my comments well.